# The Association between Working Hours Flexibility and Well-Being Prior to and during COVID-19 in South Korea

**DOI:** 10.3390/ijerph19148438

**Published:** 2022-07-11

**Authors:** Nataliya Nerobkova, Yu Shin Park, Eun-Cheol Park, Suk-Yong Jang

**Affiliations:** 1Department of Public Health, Graduate School, Yonsei University, Seoul 03722, Korea; nnerobkova@yuhs.ac (N.N.); dbtls0459@yuhs.ac (Y.S.P.); 2Institute of Health Services Research, Yonsei University, Seoul 03722, Korea; ecpark@yuhs.ac; 3Department of Preventive Medicine, Yonsei University College of Medicine, Seoul 03722, Korea

**Keywords:** flexible working hours, well-being, WHO-5, working condition survey, COVID-19

## Abstract

Objective: This study examined the relationship between the flexibility of work schedule arrangements and well-being among full-time workers prior to and after the coronavirus disease (COVID-19) outbreak in South Korea. Methods: Data from the fifth 2017 and sixth 2020–2021 Korean Working Conditions Survey, including a final sample of 45,137 participants (22,460 males; 22,677 females), were used. Multiple logistic regression was performed to establish the association between schedule arrangement types and the 5-item World Health Organization Well-Being Index. Results: The study found an association between flexible schedule arrangements and good well-being in 2017: “little flexibility” (odds ratio (OR), 1.33; 95% confidence interval (CI), 1.27–1.48), “moderate flexibility” (OR, 1.48; 95% CI, 1.28–1.71), and “high flexibility” (OR, 1.35; 95% CI, 1.06–1.72). During COVID-19, only workers with “high flexibility” were likely to have good well-being (OR, 1.49; 95% CI, 1.18–1.88), while the association between well-being and “low flexibility” (OR, 1.06; 95% CI, 0.96–1.17) and “moderate flexibility” types (OR, 0.66; 95% CI 0.59–0.75) decreased. This study found that flexible working hours may contribute to better well-being among full-time workers. However, the impact of the COVID-19 pandemic on working conditions and employee well-being should be addressed while setting working hours.

## 1. Introduction

Flexible working schedules have become an internationally widespread phenomenon [1] in all spheres of the labor market. As the demand to correspond and compete with the changing business world increases, employers must use their workforce efficiently [2]. Employees are challenged to organize their work and find a balance between work and nonwork activities [3], requiring more autonomy and an individualizing approach in the arrangement of working hours [4]. Employers have to modernize working conditions and make them more attractive, providing certain flexibility in working hours arrangements.

Findings on the effects of working time flexibility on employees’ well-being are relatively controversial, with its effects ranging from negative and neutral to positive [2]. With the ability to flexibly manage schedules, workers employ their time with better control, avoiding possible adverse outcomes [5] (e.g., overwork or burnout). Previous studies have emphasized the positive effects of flexible schedule arrangements on workers’ job satisfaction [6,7], work engagement [8], and work–life balance (WLB) [9,10].

Other findings suggest diverse effects of “company-based flexibility” and “individual-oriented flexibility,” emphasizing that the former can negatively affect employees’ general well-being [11]. Individual well-being declines when the arrangement of hours does not fit the worker’s preferred timings [12]. However, these studies are limited because of constantly evolving working conditions and various flexible working time models, making it necessary to evaluate the effect of each model accurately and decisively. Furthermore, while measured by self-reported happiness, subjective well-being may not accurately represent reality.

The coronavirus disease (COVID-19) pandemic enormously impacted the working environment worldwide, creating disruption, uncertainty, complexity, and ambiguity in all organizations [13]. Significant changes in the arrangement of working schedules and shifting to working from home via telecommuting had to be implemented. Investigators reported that work-from-home arrangements created substantial challenges for many layers of the population [14]. Thus, an ambiguous boundary between work and family emerged, particularly among employed mothers, due to childcare facilities and schools shifting to online classes during COVID-19 [14]. Employment shifts during COVID-19 also experienced changes in parent–child relational dynamics that influenced their emotional well-being negatively [15]. 

South Korea (hereafter, Korea) experienced three waves of the COVID-19 pandemic since the emergence of severe acute respiratory syndrome coronavirus 2 (SARS-CoV-2) in December 2019 [16]. The first COVID-19 case in Korea was confirmed on 20 January 2020, and the number of confirmed cases per day reached 813 on 29 February [17], which was comparatively early due to the close geographic proximity to China [14]. Although the second and third waves started with increasing social activity and poor social distancing, the spread was mitigated successfully by the rapid strengthening of social distancing measures. The COVID-19 pandemic represents a significant challenge for healthcare systems and the working population [18]. Social distancing measures to combat the spread of the virus, such as working from home and school closures, have placed a tremendous burden [19] on all sections of the population.

From May 2020, the Korean government implemented “emergency disaster relief funds” for all households regardless of income to compensate for income loss and to support local economies [14]. Given that the Korean government did not shut down businesses, the impact of COVID-19 on work environments might not be as severe as in many other countries [20]. Nevertheless, due to the pandemic, the applicant number for government unemployment benefits significantly increased by 25% since the previous year [14].

Before COVID-19, Korea was slow to implement flexible work arrangements with selective work hours and remote working. Despite governmental efforts to promote flexible work to boost employees’ well-being and enhance work–life balance, only 24.4% of workplaces had accepted flexible work arrangements in 2017 [14]. However, due to COVID-19, this number significantly increased. Regardless of the general benefits of work flexibility, the changes owing to flexible work were challenging for many workers. Flexible work arrangements allowed people to remain on the job while minimizing exposure to the virus, but it also created a great challenge to balance work and family life.

A safe and comfortable working environment must be ensured to manage occupational health effectively. The objective of the study was to examine two research hypotheses: first, if flexibility in working schedule arrangement is associated with better well-being among Korean workers, and second, if there was a difference in this association before and during the COVID-19 pandemic. A clear understanding of these relationships will provide substantial implications to policymakers and employers for considering how to best support a positive working hours experience and prevent poor mental well-being and depressive symptomology in workers.

While researchers mainly focused on examining the shifting of working sites to telecommuting and its consequences, little is known about the impact of the COVID-19 pan-demic on arrangements of working hours and whether it is related to the well-being among those workers who kept working at the working site. Another complication is that the evaluation of people’s well-being differs in every research. We used the 5-item World Health Organization Well-Being Index (WHO-5) in the current study, which is among the most widely used questionnaires for assessing subjective psychological well-being [21] and a depression predictive tool [22]. Therefore, to examine the hypotheses and fill literature gaps, we aimed to determine the association between different flexible job schedule arrangements and employees’ well-being before and during the COVID-19 pandemic, specifically among full-time workers in Korea.

## 2. Materials and Methods

### 2.1. Data Source and Sample

Data for this cross-sectional study were obtained from the fifth (2017) and sixth (2020–2021) Korean Working Conditions Survey (KWCS) conducted by the Research Institute Occupational Safety & Health Research Institute (OSHRI). The nationally representative data were collected nationwide between July and November 2017, October 2020, and April 2021 in 17 cities and provinces. The OSHRI is a one-on-one household visit interview survey conducted by professional researchers who have completed interview training. The Blaise program, a statistical survey system developed by Statistics Netherlands for a computer-assisted personal interview, was employed as a survey tool. An electronic questionnaire installed on a tablet PC was used as an alternative to the existing paper survey questionnaire. Data were collected using the paper questionnaires only when it was impossible to proceed due to a malfunction of the tablet PC [23]. Owing to COVID-19, in 2020–2021, survey data were obtained through household visit interviews using tablets, self-filling surveys (paper questionnaires), and web surveys concurrently. The survey examined the working conditions and health status of Korean workers. The KWCS has benchmarked the European Working Condition Survey research contents and methods [24] and uses a complex sampling design, including rigid multistage stratified cluster random sampling and weighted values [25]. The validity and reliability of the KWCS have been verified in previous studies [25,26]. The OSHRI is dedicated to improving the quality of life of workers and realizing social values as a public institution by conducting research based on scientific information and sharing their findings to prevent occupational diseases and accidents at industrial sites.

### 2.2. Participants

The population of the KWCS comprises all employees aged over 15 years in 50,000 households (one person per household) at the time of the survey. The survey was conducted by selecting participants considered “employed persons” within households selected as a sample group. If there was more than one eligible person, the interviews were carried out with those whose birth date was closest to the research date. All participants provided written informed consent and were guaranteed anonymity [27].

The total number of participants surveyed was 50,205 in 2017 and 50,538 in 2020–2021. We excluded individuals aged 19 years or younger, self-employed people, business owners, and unpaid family employees as their labor characteristics and environment differ from those of wage workers. Data with missing information were also excluded. Finally, data on 45,137 participants (21,494 from 2017; 23,643 from 2020–2021) were selected; full-time wage workers comprised the representative sample. Because the OSHRI is a secondary dataset available in the public domain without identifiable information, our study did not require approval from the institutional review board or informed consent.

### 2.3. Variables

The variable of interest was “job schedule arrangement flexibility.” We assessed this based on the question, “How are your working time arrangements set?” Workers were offered four possible options: (1) they are set by company/organization with no possibility of changes—“no flexibility”; (2) you can choose between several fixed working schedules —“little flexibility”; (3) you can adapt your working hours within certain limits—“moderate flexibility”; and (4) your working hours are entirely determined by yourself—“high flexibility”.

The outcome variable was identified using WHO-5, a short and generic global rating scale that measures subjective well-being. Since its first publication in 1998, the WHO-5 has been translated into more than 30 languages and used in research studies worldwide, proving its reliability and validity [21,28,29,30,31]. In Korea, the inter-rater reliability of WHO-5 in a previous study was 0.80 [32]. WHO-5 contains only positively phrased items. Each of the five items was scored from 5 (all of the time) to 0 (at no time). The items are (1) “I have felt cheerful and in good spirits”, (2) “I have felt calm and relaxed”, and (3) “I have felt active and vigorous”, (4) “I woke up feeling fresh and rested”, and (5) “My daily life has been filled with things that interest me”. The instruction is, “Please indicate for each of the five statements that are the closest to how you have been feeling over the last two weeks”. The final score is calculated by adding the scores of 0, 1, 2, 3, 4, and 5 for the five statements. The total raw score, ranging from 0 to 25, is multiplied by 4 to give the final score, with 0 representing the worst imaginable well-being and 100 representing the best imaginable well-being [22]. Notably, the layout of the WHO-5 follows that of the Major Depression (ICD-10) Inventory, which measures WHO/ICD-10 symptoms of depression [22]. WHO recommends administering the ICD-10 inventory if the raw score of WHO-5 is below 13 or if a person answers 0 to 1 for any of the five items. Previous studies emphasized that WHO-5 has enough discriminatory validity as a screening tool for the detection of depressive episodes. Its standard cut-off point of <13 has a significant sensitivity/specificity trade-off [28]. Thus, the cut-off point for poor well-being was taken as a score below 13 or a score of 0 or 1 in at least one of the statements. Good well-being was considered in those whose score was more than or equal to 13. For sensitivity analyses, we also considered scores of ≥7 and ≥19.

The study includes three working characteristic variables. The weekly work hours were classified as ≤40, 41–52, and >52 h according to the *Labor Standards Act 2018*. Commuting time was categorized into less than 30 min, from 30 to 60 min, and more than 60 min, covering a round trip per day. Finally, monthly income amount was divided into four quartiles (<1,800,000, <2,500,000, <3,500,000, ≥3,500,000 in 2017, and (<2,000,000, <2,500,000, <3,500,000, ≥3,500,000 in 2020–2021). Further, KRW 1304 is equivalent to USD 1 (as of 26 June 2022). We also included the following sociodemographic variables: gender (men, women), age (20–29 years old, 30–39 years old, 40–49 years old, 50–59 years old, and 60 years old or over), and educational background (high school or less, and undergraduate or more). Socioeconomic factor included subjective health perception (good, poor).

### 2.4. Statistical Analysis

A descriptive analysis was completed to explore the distribution of the general characteristics of the study population. A chi-square test investigated the association between job schedule flexibility and WHO-5 among workers. A multiple logistic regression analysis was employed to evaluate this relationship and perform subgroup analyses stratified by sex, age, educational level, income per month, working hours per week, commuting time, and self-perceived health status. The results were reported as odds ratios (ORs) and 95% confidence intervals (CIs). Model fitting was performed after using the PROC SUR-VEYLOGISTIC procedure. There was found no multicollinearity using the variance inflation factor. All data analyses were conducted using the SAS 9.4 software (version 9.4; SAS Institute Inc., Cary, NC, USA). A weighted logistic regression procedure was used to account for the compound and stratified sampling design. A two-sided *p*-value of <0.05 was considered to indicate statistical significance.

## 3. Results

Table 1 summarizes the general characteristics of the study population. Of the total 45,137 participants, 6466 (30.1%) people in 2017 and 8630 (36.5%) people in 2020–2021 received a WHO-5 score below 13, indicating their poor well-being. Interestingly, in 2020–2021, in the case of moderately flexible job arrangements, there was a noticeable increase of 20% in workers who received low WHO-5 scores.

Table 2 depicts the findings of the logistic regression analysis stratified by 2017 and 2020–2021 and the association between the four types of job schedule arrangement flexibilities and WHO-5 scores over 13. The findings suggested that compared with the flexible schedule arrangement, in 2017, there was an association between all flexible types of job arrangements and higher well-being scores, including “little flexibility” (OR, 1.33; 95% CI, 1.21–1.46), “moderate flexibility” (OR, 1.48; 95% CI, 1.28–1.71), and “high flexibility” (OR, 1.35; 95% CI, 1.06–1.72). Notably, in 2020–2021, only workers with “high flexibility” job schedules were associated with higher WHO-5 scores (OR, 1.49; 95% CI, 1.18–1.88), while the association with “moderate flexibility” dropped significantly (OR, 0.66; 95% CI, 0.59–0.75).

The independent subgroup analysis findings of the variables associated with WHO-5 scores over 13 and schedule arrangement flexibility are shown in Figure 1. The difference in the significance of this association before and during the pandemic may be detected. Therefore, in 2017, workers with little or moderate flexibility in schedule arrangements reported good well-being. In contrast, 2020–2021 revealed that only workers with highly flexible working hours tended to have good well-being.

Figure 2 summarizes the sensitivity analysis showing the results of different cut-off points (WHO-5 scores of 7 or more and 19 or more) sorted by the variable of interest (no flexibility, little flexibility, moderate flexibility, and high flexibility) in 2017 and 2020–2021. In 2017, there was a consistent association between WHO-5 score and schedule arrangement types: score over 7 (OR, 1.16; 95% CI, 1.02–1.32) and score over 19 (OR, 1.17; 95% CI, 1.08–1.26). The results were identical in 2020–2021; however, only in a “high flexibility group” supporting the primary analyses outcomes: score over 7 (OR, 1.43; 95% CI, 1.13–1.81) and score over 19 (OR, 1.55; 95% CI, 1.23–1.95) (the *p*-value is <0.0001).

## 4. Discussion

Work schedule arrangement flexibility affects employees’ well-being to a particular extent. The power of this impact may vary based on the flexibility type. To this end, we evaluated four leading types of schedule organization—no flexibility, low flexibility with a choice between several fixed working schedules determined by the organization, moderate flexibility with the possibility of adapting working hours within certain limits (e.g., flextime), and full flexibility when working hours are entirely determined by employees—with respect to well-being using the WHO-5 in South Korea before and after the COVID-19 outbreak by using nationally representative survey data, which provided detailed evidence to support this relationship.

Therefore, the primary outcomes of the current research suggest that workers who had certain flexibility in arranging their job schedules were more likely to have a higher well-being score than those without any flexibility (where working times are set by the organization with no possibility for changes). Associations were found between each of the flexible arrangement types mentioned in this study; however, differences appeared in the prominence of this correlation. Therefore, in 2017, before the COVID-19 outbreak, low flexibility and full flexibility showed nearly identical associations with good well-being. In comparison, moderate flexibility (e.g., flextime) was associated more significantly with a WHO-5 score over 13.

However, the results from 2020–2021 after the COVID-19 outbreak indicated that employees who had full flexibility in arranging their work schedules showed an association with good well-being. In contrast, the association of good well-being with moderate and little flexibility decreased. Notably, the total number of people with well-being scores below 13 increased after the COVID-19 outbreak, which can be explained by the difficulties and restrictions it applied to the population worldwide [33,34].

The mechanism that may explain the reason behind such a sharp alteration in the results after the COVID-19 outbreak may lie in the changed workplace conditions and working environment due to the actions taken by businesses and restrictions implemented by the government to curb the spread of COVID-19. The COVID-19 pandemic has created a dilemma between economic stimuli and public health control [16]. The findings revealed several occupational risk factors that challenged healthcare workers at high risk of mental health outcomes, including burnout syndrome [35]. For instance, wherever possible, people are now working from home with adapted working hours, including possible interruptions and distractions (e.g., due to family obligations) [36]. Although some people value this flexibility, reports also suggest that these shifts led to increased daily working hours [37], accumulating work stress, fatigue, and time conflict, resulting in additional risks of illness, injury, burnout, job dissatisfaction, and WLB instability [12].

Researchers [36] underlined that the COVID-19 pandemic and the accompanying measures could affect several work- and career-related variables: (1) working conditions, (2) work motivations and behavior, (3) job and career attitudes, (4) career development, and (5) personal health and well-being. COVID-19 is primarily responsible for the change in traditional fixed-hour work [38]. While workers whose working hours are flexible are naturally limited, flextime might be beneficial to both employees and employers. A highly committed and well-resourced employer might overcome the structural obstacles posed by flextime [38]. However, flextime was ruled out for a considerable fraction of the workforce. The more comprehensive the range of working hours, the less time employees spend together, which imposes restrictions on team meetings and the resolution of general issues, which is crucial during the COVID-19 pandemic.

Comparing our study’s findings has proved to be complicated. While the association between flexible working hours and well-being has been thoroughly investigated in the pre-COVID-19 era, there is a lack of studies comparing the association between two time periods, before and during the COVID-19 pandemic. Nevertheless, one study in Korea examined how remote working from home during the pandemic was associated with well-being. It was suggested that parents who worked at home struggled to combine work and parenting, as their children also stayed home due to closed daycare facilities and schools [14]. Another study emphasized how “smart working” and teleworking helped facilitate management and work autonomy and reduce stress levels and pressures experienced in the workplace. However, technostress for the necessity of learning new procedures and modifying their work routines owing to COVID-19 became more acute [39].

This study had several strengths and limitations. The strength of this study is that we used the most current data available on the analyzed variables. The study may provide an in-depth view of the association between flexibility in working hours and employees’ well-being prior to and during the COVID-19 outbreak. Additionally, since we used a standardized tool for measuring mental well-being, the findings provide a substantial basis for future studies.

The limitation of this study is that it was based on data from a cross-sectional study. Therefore, although associations were found between several variables, causal order could not be determined. Longitudinal and quasi-experimental studies are urgently needed for further validation of the results. Second, this study relied on self-reported data. Hence, people may not have accurately reported on their working conditions and well-being. Third, as the cut-off points for the variables were adopted from the OSHRI, it may be difficult to generalize our findings to different settings or populations. Future data and analysis are required to fully discern the link between specific work schedule arrangement flexibility types and employee well-being. Finally, it is important to note that these data were obtained within the early period of the COVID-19 pandemic in Korea. Therefore, the findings may not represent the workers’ well-being after the restrictions and changes in the working environment became increasingly routine.

## 5. Conclusions

The study presented the results of two working condition surveys regarding work time flexibility and its association with workers’ well-being before and during the COVID-19 pandemic. Workers with flexible arrangements tended to have better well-being than those with nonflexible arrangements. However, COVID-19 has affected employees’ conditions, resulting in poor well-being even under flexible schedules. However, high flexibility seemed to alleviate the adverse effect of the pandemic.

This study highlights factors that impact workers’ mental health well-being while having flexible working hours and provides a foundation for investigating how to best support a positive working hours experience. Therefore, the impact of COVID-19 on the working environment and employees’ well-being should be addressed while setting working hours.

## Figures and Tables

**Figure 1 ijerph-19-08438-f001:**
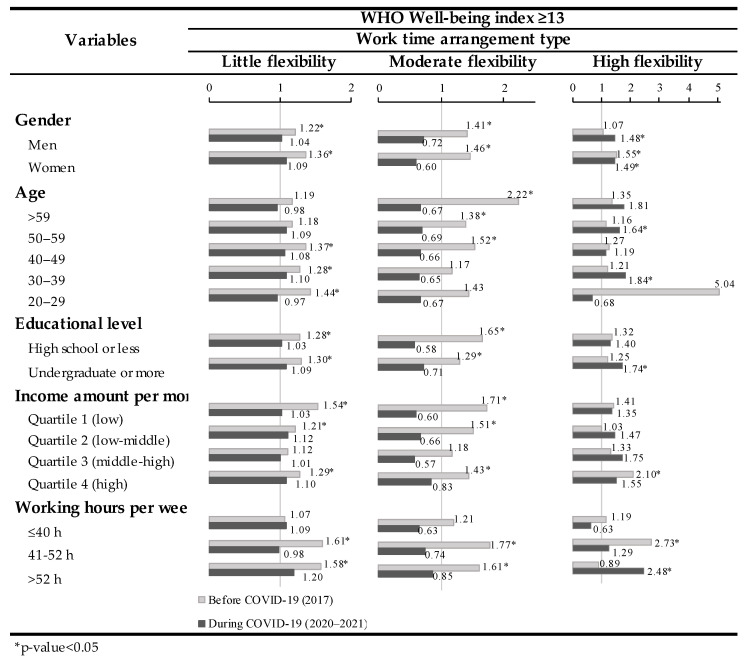
The results of subgroup analysis stratified by independent variables and year.

**Figure 2 ijerph-19-08438-f002:**
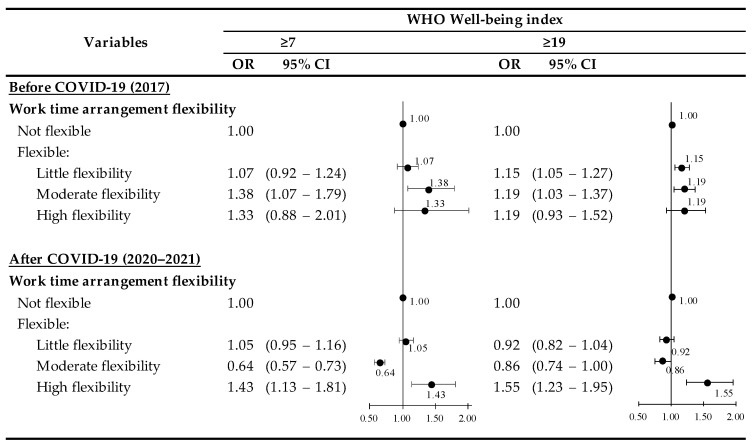
The results of sensitive analysis stratified by work time arrangement type and index of well-being.

**Table 1 ijerph-19-08438-t001:** General characteristics of the study population.

Variables	WHO Well-Being Index-5											
	Before COVID-19 (2017)						During COVID-19 (2020–2021)					
	TOTAL		Bad (Index < 13)		Good (Index ≥ 13)		TOTAL		Bad (Index < 13)		Good (Index ≥ 13)	
	N	%	N	%	N	%	N	%	N	%	N	%
(N = 45,137)	21,494	100.0	6466	30.1	15,028	69.9	23,643	100.0	8630	36.5	15,013	63.5
Work time arrangement type												
Not flexible *	17,292	80.5	5364	31.0	11,928	69.0	20,300	85.9	7322	36.1	12,978	63.9
Flexible:												
Little flexibility **	2747	12.8	740	26.9	2007	73.1	1851	7.8	677	36.6	1174	63.4
Moderate flexibility ***	1101	5.1	265	24.1	836	75.9	1125	4.8	520	46.2	605	53.8
High flexibility ****	354	1.6	97	27.4	257	72.6	367	1.6	111	30.2	256	69.8
Gender												
Men	10,892	50.7	3268	30.0	7624	70.0	11,568	48.9	4302	37.2	7266	62.8
Women	10,602	49.3	3198	30.2	7404	69.8	12,075	51.1	4328	35.8	7747	64.2
Age												
>59	1615	7.5	687	42.5	928	57.5	2691	11.4	1195	44.4	1496	55.6
50–59	4527	21.1	1466	32.4	3061	67.6	5624	23.8	2127	37.8	3497	62.2
40–49	6223	29.0	1870	30.0	4353	70.0	6429	27.2	2367	36.8	4062	63.2
30–39	5823	27.1	1642	28.2	4181	71.8	5973	25.3	2011	33.7	3962	66.3
20–29	3306	15.4	801	24.2	2505	75.8	2926	12.4	930	31.8	1996	68.2
Educational level												
High school or less	8161	38.0	2940	36.0	5221	64.0	8795	37.2	3687	41.9	5108	58.1
Undergraduate or more	13,333	62.0	3526	26.4	9807	73.6	14,848	62.8	4943	33.3	9905	66.7
Income amount per month												
Quartile 1 (low)	6021	28.0	2077	34.5	3944	65.5	6728	28.5	2770	41.2	3958	58.8
Quartile 2 (low–middle)	6075	28.3	1800	29.6	4275	70.4	5595	23.7	2105	37.6	3490	62.4
Quartile 3 (middle–high)	5629	26.2	1573	27.9	4056	72.1	6713	28.4	2233	33.3	4480	66.7
Quartile 4 (high)	3769	17.5	1016	27.0	2753	73.0	4607	19.5	1522	33.0	3085	67.0
Working hours per week												
≤40 h	11,996	55.8	3315	27.6	8681	72.4	16,422	69.5	5770	35.1	10,652	64.9
41–52 h	6412	29.8	1948	30.4	4464	69.6	5666	24.0	2172	38.3	3494	61.7
>52 h	3086	14.4	1203	39.0	1883	61.0	1555	6.6	688	44.2	867	55.8
Commuting time												
>60 min	2588	12.0	850	32.8	1738	67.2	3824	16.2	1470	38.4	2354	61.6
30–60 min	9264	43.1	2660	28.7	6604	71.3	9608	40.6	3555	37.0	6053	63.0
<30 min	9642	44.9	2956	30.7	6686	69.3	10,118	42.8	3562	35.2	6556	64.8
Self-perceived health status												
Good	16,288	75.8	4127	25.3	12,161	74.7	17,452	73.8	5334	30.6	12,118	69.4
Poor	5206	24.2	2339	44.9	2867	55.1	6191	26.2	3296	53.2	2895	46.8

* Working time is set by a company/organization with no possibility for changes; ** choice between several fixed working schedules determined by a company or an organization; *** possible adaptation of working hours within certain limits (e.g., flextime); **** working hours are entirely determined by a worker.

**Table 2 ijerph-19-08438-t002:** Results of factors associated with a high well-being index.

Variables	WHO Well-Being Index ≥ 13			
	Before COVID-19 (2017)		During COVID-19 (2020–2021)	
	OR	95% CI	OR	95% CI
Work time arrangement flexibility				
Not flexible	1.00		1.00	
Flexible:				
Little flexibility	1.33	(1.21–1.46)	1.06	(0.96–1.17)
Moderate flexibility	1.48	(1.28–1.71)	0.66	(0.59–0.75)
High flexibility	1.35	(1.06–1.72)	1.49	(1.18–1.88)

Adjusted for sex, age, educational level, income per month, working hours per week, commuting time, and self-perceived health status.

## Data Availability

The Korean version of the Working Conditions Survey (KWCS) is publicly available at http://www.kosha.or.kr/ (accessed on 23 April 2022) under permission from the Korea Occupational Safety and Health Agency (KOSHA).

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
