# Peer review of "The Association between Working Hours Flexibility and Well-Being Prior to and during COVID-19 in South Korea"

_ijerph, 2022, doi:10.3390/ijerph19148438_

Round 1

Reviewer 1 Report

The submitted paper is well organized and prepared. The topic is of interest for the readership of the Journal. The samples used in the study are representative and the obtained results may be considered relevant. No ethical issues have been observed.

There are some recommendations for improvement that are discussed below. They should be implemented before considering the paper for publication.

The analysis results are presented in a tabular form only. It is highly recommended to produce various graphs in order to help a better grasp. Humans are rather visual.

Limits of the approach and study should be discussed and future works defined.

Conclusions are rather modest, and should be further elaborated.

The references should be written as per Journal rules.

Author Response

We were pleased to have the opportunity to revise our paper. In revising our paper, we have carefully considered your comments and suggestions. As instructed, we have attempted to explain the changes made in reaction to all of the reviewers’ comments. The reviewers’ comments were very helpful overall, and we appreciate the constructive feedback on our original submission. After addressing the issues raised, we feel the quality of the paper has greatly improved, and we hope you agree. Our response to each comment is as follows, and we attach a revision note with the highlighted, revised sections of the manuscript. Again, thank you for the valuable and helpful comments.

Reviewer 2 Report

ID: ijerph-1765665

Title: The association between working hours flexibility and well-being prior to and during the COVID-19 in South Korea.

Thank you for providing a chance to review this manuscript.

Comment: Major revision.

Detailed information:

Abstract

Abstract, page 1: What’s the interval range of your CI? 90%? 95%? The logic of the abstract is clear, but I recommend breaking it down into “Objectives”, “Methods”, “Results”, and “Conclusions” to make the information more accessible to the reader.

1. Introduction

Paragraph 3, page 1-2: Since it has been mentioned here that “subjective well-being may not accurately represent reality”, did your research design take them into account to control the part of the error?

Paragraph 2, page 2: Given that the participants in this study are all Korean, can you describe more about how the work environment in Korea has been affected?

2. Materials and Methods

2.1. Data source and sample

Paragraph 3, page 2: How did you conduct a survey “using a tablet over an existing paper questionnaire”? How did you perform the quality control?

2.2. Participants

Paragraph 1, page 2: "Employee" is not clearly defined in this study. And does this create more bias?

2.3. Variables

Paragraph 1, page 3: These so-called variables of interest you measured, are they part of a scale? If yes, have they been validated? If not, how did you ensure the reliability and validity of the measurement?

Paragraph 2, page 3: Glad to see the detailed information on the WHO-5, but what are the measurement properties of it in the current study? And the full name of WHO-5 has appeared, and it is better to directly use the abbreviation here.

Paragraph 3, page 3: Any evidence for the category of these working characteristic variables?

Paragraph 4, page 3: Adjusted? What does this mean? Did you mean to adjust variables for your regression model? If yes, it should not be written here but in the “Statistical Analysis” part.

2.4. Statistical Analysis

Paragraph 5, page 3: I do not think the description of the Statistical Analysis is adequate. Expand more information on your statistical analysis for the readers to understand.

3. Results

Paragraph 1, page 4:The total percentage of people with WHO-5 scores below 13 rose by 6.4% in 2020-2021 compared with 2017”: Not an easy sentence to understand, rephrase it please.

Paragraph 2, page 4: Even though you have mentioned the CI was an interval of 95% in your “Method” part, we still need to represent it as “95% CI” in every place it appears. Check other same problems in your article too. Moreover, check the use of your punctuations (e.g., the use of brackets).

Paragraph 3, page 4: Very unclear writing of this paragraph. You may need to rewrite it.

All the presents of tables are too rough to read, format all of them.

4. Discussion

If you intended to add line numbers, you add them to all your text. And your page numbers are such a mess!

Paragraph 4, page 11: Have there been similar studies before? If so, what are the similarities and differences with the results of your study? In addition, please show the innovation and practical implications of the study. And part of the "Discussion" can be put into the "Introduction" to enrich the significance and innovation of this research.

With all due respect, I did not even finish all the reading. Some of your expression contains too much redundancy that makes people have no intention of reading. And the sudden appearance of line numbers and messed format make this “manuscript” more of a DRAFT to me. It’s unacceptable.

First, reading more papers from the TOP journals, to learn the formats, expressions, and of great importance—logic, might help a lot before revising. Second, rephrase your sentences to make your expressions clear. Some of your sentences and paragraphs are hard to read and understand. Third, accordingly arrange your tables and results. Do not let the readers do your job. Last, finding a native English speaker to improve the writing can considerably improve the quality.

Thank you and my best,

Your reviewer

Author Response

(The authors gave the same response as above.)

Reviewer 3 Report

Dear authors,

Thank you for the opportunity to read your work.

I hope I can offer some “food for thought” for your work.

The dataset used is strong and interesting.

From my point of view, the abstract is correct, although if you could, add some information about the flexible working schedule volition. What I’m saying is: this flexible working schedule is for everybody or only for those who ask? In other words, is it an HR practice or an idiosyncratic deal? In this case, was it controlled by self-employed and company-employed? It may change.

Another point is about time flexibility, it is only possible when a service encounter is not expected in the sense that a teacher needs to be there on the time agreed. This flexibility is not possible for all kinds of jobs. Thus, it could be described at the beginning of what type of job you are talking about. My point here is that asking only how flexible is the job does not answer an important question: if the person expected as flexible. Sometimes it is very good to have a working schedule because when the work finish, it finishes, it does not “cross the line” of your home. What I mean is that it may depend on the person, family and expectations. 

The authors also described the gap in the literature regarding flexible working arrangements and their impact on wellbeing. However, as the authors described it is important to control family arrangements, more specifically, if the family needs to take care of others (elders or youngers), the type of job and the type of contract (or its lack of contract).

Another interesting point that could be added is a description of working arrangements in Korea, in the sense of how many hours is a “common” week in South Korea? 32h? 40? 44h? Other information in Galloup of wellbeing, how many unemployed or vulnerable there are, ourworldindata graph of covid waves could be included as well to explain the situation to the readers. 

Method:

Why did the authors decide to reduce the flexibility categories as described here: “For the subgroup analysis, we also grouped these types into two categories: “not flexible” and “flexible” job schedule arrangements.” and after the authors split again into sub-categories. 

It is not clear what are the real aim of the work and why the authors do not use the linearity of the variables. Why are you reducing or aggregating it if you could conduct a linear regression instead of a multinomial regression? It is not clear the point.

For example, the aim of the work: “examined the relationship between the flexibility of work schedule arrangement and well-being among full-time workers prior to and after the coronavirus disease (COVID- 19) outbreak in South Korea. ”. What is the aim of the work? Why authors are examining the relationship between flexibility of work schedule arrangement? to predict and avoid depression? Although it is not assumed, it is used as a cut-off criterion. The reliability of well-being was not described.

The point is that there are many analyses that are not clear what is its aim or if there is any hypothesis associated with them that is not clear to the reader. To achieve the objective of the work you need to conduct regression and a multi-group comparison (prior and after Covid-19). Another possible point, that is also not clear, is if the authors are interested in personal variables (i.e., gender, age, educational level, working hours per week) as variables that could moderate the flexibility-wellbeing relationship.

The work has many strengths, particularly being a powerful sample and data. However, it is needed to clarify the aim and pursue it. I suggest adjusting in part the introduction of the work introducing the hypothesis and rewriting a little the objective in order to clarify the contribution of the text. 

Again, thank you very much for the opportunity.

Author Response

(The authors gave the same response as above.)

Round 2

Reviewer 2 Report

The authors have made substantial improvement and have addressed the comments. The paper showed better clarity and an improved analysis of data.